# Low-Light Image Enhancement Using Hybrid Deep-Learning and Mixed-Norm Loss Functions

**DOI:** 10.3390/s22186904

**Published:** 2022-09-13

**Authors:** JongGeun Oh, Min-Cheol Hong

**Affiliations:** School of Electronic Engineering, Soongsil University, Seoul 156-743, Korea

**Keywords:** low-light image enhancement, hybrid deep-learning, mixed-norm, halo artifact, color distortion

## Abstract

This study introduces a low-light image enhancement method using a hybrid deep-learning network and mixed-norm loss functions, in which the network consists of a decomposition-net, illuminance enhance-net, and chroma-net. To consider the correlation between R, G, and B channels, YCbCr channels converted from the RGB channels are used for training and restoration processes. With the luminance, the decomposition-net aims to decouple the reflectance and illuminance and to train the reflectance, leading to a more accurate feature map with noise reduction. The illumination enhance-net connected to the decomposition-net is used to enhance the illumination such that the illuminance is improved with reduced halo artifacts. In addition, the chroma-net is independently used to reduce color distortion. Moreover, a mixed-norm loss function used in the training process of each network is described to increase the stability and remove blurring in the reconstructed image by reflecting the properties of reflectance, illuminance, and chroma. The experimental results demonstrate that the proposed method leads to promising subjective and objective improvements over state-of-the-art deep-learning methods.

## 1. Introduction

The enhancement and miniaturization of image sensors make it possible to easily obtain high-quality images. However, it still remains challenging to overcome external environmental factors, which are the main causes of image degradation and distortion. One such factor, low light, can be a bottleneck in the use of captured images in various applications, such as monitoring, recognition, and autonomous systems [1,2]. A low-light image can be enhanced by adjusting the sensitivity and exposure time of the camera; however, it leads to a blurred image.

Many approaches have been exploited to enhance images in recent decades. Low-light image enhancement can generally be classified into contrast ratio improvement, brightness correction, and cognitive modeling methods. Histogram equalization has been widely used to improve the contrast ratio, and gamma correction has been used to improve image brightness information. However, these methods have limitations in performance improvement because they use arithmetic or statistical methods without considering the illuminance component of an image. Cognitive modeling-based methods correct low illuminance and distorted color signals by dividing the acquired image into illuminance and reflectance components using retinex theory [3]. Single-scale retinex (SSR) [4] and multi-scale retinex (MSR) [5,6] methods have been used to reconstruct low-light images based on retinex theory and random spray [7,8] and illuminance model-based methods [9,10,11,12] have been developed as modified versions. Because methods based on the retinex model improve the image by estimating the reflection component, there are problems that cause halo artifacts and color distortion [13]. In addition, variational approaches using optimization techniques have been proposed, but their performance depends on the choice of parameters, and the computational cost is very high [14,15,16]. Recently, deep-learning-based image processing research has been actively conducted, and various deep-learning methods have been exploited to enhance or reconstruct low-light images [17,18,19,20,21,22].

Deep-learning-based low-light image restoration methods have advantages and disadvantages depending on their structural characteristics [2]. Most deep-learning methods apply the same architecture to the RGB channels. However, it has been shown that the correlation between the R, G, and B channels is very low; therefore, different architectures suitable to each channel or different color spaces would be more desirable to obtain more satisfactory results. In addition, deep-learning approaches based on the retinex model have been exploited to enhance low-light images. Most of them aim to decouple the reflectance and illuminance components from an input image and enhance only the reflectance [20]. For example, MSR-net using a one-way convolutional neural network (CNN) structure results in color distortion [18]. Retinex-net uses a decomposition neural network (DNN) to decouple the reflectance and illuminance conforming to the retinex-model. However, without considering the different characteristics of the RGB channels, each channel is learned through the same structure, resulting in unstable performance and halo distortion. MBLLEN [21] and KIND [22] attempt to simultaneously control low illuminance and blur distortion using an auto-encoder structure. However, they lead to a loss of detailed image information. Recently, unsupervised learning methods have been reported to solve an over-fitting problem of deep-learning networks on paired images. For example, EnlightGAN uses generator and discriminator models to consider a more realistic environment. Although it leads to promising results, it is tough to train the two models at the same time [23]. In addition, Zero-DCE uses incisive and nonlinear curve mapping [24]. However, unsupervised learning methods have a limitation on performance because a reference image is not used in the loss function. 

As described above, deep-learning-based low-light image restoration methods have the problems such as (1) color distortion due to insufficient correlation between color channels and (2) unstable performance and distortion due to the use of the same color channel structure.

To address the above problems, the reflectance and illuminance components are decoupled from the luminance channel of the YCbCr space in this study because the luminance histogram is more similar to the brightness histogram and chrominance channels are less sensitive to additive noise. Based on the converted YCbCr channels, we propose a hybrid structure using decomposition-net, illuminance enhance-net, and chroma-net. The decomposition-net decouples the reflectance and illuminance with the reduction in the additive and shares the weight by extracting the feature map. The illuminance enhance-net connected to the decomposition-net is used to enhance the decoupled illuminance by reducing the halo artifact, which is the main distortion of the retinex-based approaches. In addition, a chroma-net is independently utilized to enhance chroma signals by minimizing color distortion. Moreover, a mixed norm loss function used in each training net is introduced to minimize the instability and degradation of the reconstructed images by reflecting the properties of the reflectance, illuminance, and chroma. The performance of the proposed method is validated using various quantitative evaluators.

The remainder of the paper is organized as follows. Section 2 introduces the proposed deep learning structure and mixed norm-based loss function for low-light image reconstruction. The experimental results and analysis are described in Section 3, and the conclusions are presented in Section 4.

## 2. Proposed Method

### 2.1. Hybrid Deep-Learning Structure

The retinex model is the most representative cognitive model for low-light image enhancement, and it can be expressed as follows [3]:(1)S=R·L,
where *S*, *R*, and *L* represent the perceptual scene (intensity) of the human eye, reflectance, and illuminance, respectively. Equation (1) is a model that experimentally demonstrates that an object’s color varies with ambient illuminance in the human visual system.

This study introduces a hybrid neural network to simultaneously improve illuminance and reflectance components. As mentioned, most deep-learning networks based on the conventional retinex model lead to halo artifacts because they aim to enhance only the reflectance component by decomposing the reflectance and illuminance components from an observed low-light image. Additionally, many deep-learning methods based on the retinex model suffer from color distortion owing to the lack of consideration of the correlation between color channels [25]. To solve these problems, this study adopts a decomposition network that decouples the illuminance and reflectance components and enhances the reflectance in the YCbCr color space. It has been demonstrated that luminance is highly effective in estimating illuminance [26]. Accordingly, illuminance and reflectance are decomposed from the luminance channel, and each channel is used in the training process. In this study, the luminance channel in the YCbCr space is used as an input of the decomposition-net, such that Equation (1) can be rewritten as follows:(2)y=logY=r+l=logR+logL,
where Y represents the luminance channel of an observed low-light image, and R and L denote the reflectance and illuminance components of the Y channel, respectively. In addition, the illuminance enhance-net and chroma-net are considered to enhance each component in this study. 

The deep-learning structure based on the retinex model should be able to effectively reflect the characteristics of the illumination and reflectance components. In particular, the local homogeneity and spatial smoothness of the illumination component should be effectively decomposed, and the local correlation of the reflectance component should be efficiently extracted [27,28,29]. Additionally, it is desirable for the network to be capable of removing additive noise. 

Figure 1 shows a conceptual diagram of the proposed decomposition network. As shown in the figure, the reflectance and illuminance components decoupled from the luminance channel share weights by extracting feature maps that conform to the model by specifying a loss function for each output. In Figure 1, $y_{low}$, ${\bar{l}}_{low}$, and ${\bar{r}}_{low}$ represent the low-light luminance, trained illuminance, and trained reflectance components, respectively. In contrast, $y_{GT}$, ${\bar{l}}_{GT}$, and ${\bar{r}}_{GT}$ represent the paired ground-truth luminance, trained illuminance, and trained reflectance, respectively.

As shown in Figure 2, the proposed deep-learning network consists of three stages: (1) decomposition-net, (2) illuminance enhance-net, and (3) chroma-net. As previously mentioned, the decomposition-net accurately decouples the reflectance and illuminance components from the luminance channel. In addition, the illuminance enhance-net is used to learn about the illuminance, and chroma-net is added to consider the chroma characteristics.

The proposed decomposition-net considers the characteristics of each component and composites the sub-network structure to facilitate training. Sub-neural networks consist of a forward CNN, an auto encoder-based neural network, and a multi-scale-based neural network using skip-connections. For illuminance, a multi-scale CNN structure including various-sized receptive fields is used to decompose the local homogeneity and spatial smoothness. This structure is capable of obtaining a feature extraction map that is robust to various input images [2]. The reflectance component is easier to preserve and learn detailed information and boundary information of the image than the illumination component. Applying these characteristics, the reflectance component uses a forward small-scale receptive field to facilitate learning the local correlation of the image. The auto-encoder has a structure that combines learning feature maps of different sizes using a skip- connection. It has been shown that this structure is easy to learn through structural analysis of the image and that it is effective in removing additive noise in the image [30]. As described above, to effectively remove the noise present in the low-light image, an auto-encoder structure is used for the reflection component. Figure 3 shows the structural diagrams for the multi-scale CNN (sub-net 1), forward CNN (sub-net 2), and auto-encoder (sub-net 3) used in this study.

The parallel structure described above becomes structurally flexible by learning the decomposition components, thereby shortening the learning time and clarifying the role of each sub-network. In addition, illuminance enhance-net and chroma-net use a forward CNN to enhance each component because illuminance and chroma include less additive noise than reflectance, such that over-blurring can be avoided. 

### 2.2. Mixed Norm-Based Loss Function

A loss function using the hybrid learning architecture is defined to effectively train the input pair by minimizing the error of each learning system, i.e., the decomposition-net, illuminance enhance-net, and chroma-net.

The decomposition-net accurately extracts the reflectance from the luminance, and the loss function is defined as follows:(3)$$Loss^{D}=L_{d}+L_{r}+L_{l},$$
where $L_{d}$, $L_{r}$, and $L_{l}$ represent the decomposition, reflectance, and illuminance loss functions, respectively. 

The decomposition loss function is a basic loss function using the retinex model and can be written as follows:(4)$$L_{d}=\|{\bar{r}}_{low}-{\bar{r}}_{GT}{\|}_{1}+\alpha_{1}\|{\bar{r}}_{low}+{\bar{l}}_{GT}-{\tilde{y}}_{GT}{\|}_{2}^{2}+\alpha_{2}\|{\bar{r}}_{low}+{\bar{l}}_{low}-{\tilde{y}}_{low}{\|}_{2}^{2},$$
where ${\tilde{y}}_{GT}$ and ${\tilde{y}}_{low}$ denote the normalized ground truth luminance channel and paired low-light luminance, respectively, in which the elements of $\;{\tilde{y}}_{GT}$ and ${\tilde{y}}_{low}$ are scaled to [0, 1]. For an M×N-sized image, each symbol is a lexicographically ordered MN×1 column vector. In Equation (4), each term represents the model-based loss function in which the first term uses the L1 norm because it includes the details and boundary information of the image. 

The reflectance model loss function should contain detailed information regarding the object. Therefore, the minimization terms for the gradient map and the error term for the ground-truth image are included to reflect the property. The loss function for the reflection model is expressed as follows:(5)$$L_{r}=\beta_{1}\|\nabla {\bar{r}}_{low}-\nabla {\tilde{y}}_{GT}{\|}_{1}+\beta_{2}\|{\bar{r}}_{low}-{\tilde{y}}_{GT}{\|}_{2}^{2},$$
where ∇ represents the gradient operator, and $\beta_{1}$ and $\beta_{2}$ denote the regularization parameters to control the relative contribution of each term.

In general, illuminance is suitable for representing an object surface as a Lambertian model [14]. Accordingly, the illuminance model loss function can be expressed as follows:(6)$$L_{l}=\gamma \left[\frac{\nabla {\bar{l}}_{low}}{max\left(\nabla {\tilde{y}}_{low},\varepsilon \right)}+\frac{\nabla {\bar{l}}_{GT}}{max\left(\nabla {\tilde{y}}_{GT},\varepsilon \right)}\right],$$
where γ is the regularization parameter for the loss function, and ε is a small constant to prevent the denominator from becoming zero.

The loss functions for training the illuminance component and chroma signals are expressed as follows:(7)$$Loss^{L}=\|{\bar{l}}_{row}^{enh}-{\bar{l}}_{GT}{\|}_{2}^{2},$$
where the initial vector is equal to ${\bar{l}}_{low}$, and
(8)$$Loss^{C}=\|{\bar{C}}_{row,i}-{\tilde{C}}_{GT,i}{\|}_{2}^{2}\;i\in \left\{r,b\right\},$$
where *i* denotes the chrominance channel index and ${\tilde{C}}_{GT,i}$ represents the normalized *i*-th chrominance signal of the ground-truth image. An element of the chrominance takes a value between −128 and 128; thus, it is normalized to [0, 1] for training. The Adam method [31] was used to obtain the optimized solution of the loss functions, and 55 batch sizes were applied in the training process.

As expressed above, the proposed decomposition loss function consists of various functions; therefore, the convergence of the loss function depends on the choice of the parameters. The selection of optimized parameters is beyond the scope of this work. In this study, these parameters were experimentally determined. The regularization parameters ($\alpha_{1}$ and $\alpha_{2}$) in Equation (4) are due to the retinex theory, and when they have low values, the decomposition-net fails to extract an accurate feature map. It was observed that the decomposition-net satisfactorily converged with $\alpha_{1},\;\alpha_{2}>0.1$. In addition, it was confirmed that as $\beta_{1}$ in Equation (5) increases, detailed information, such as boundaries, is well expressed in the feature map of the reflectance component. However, it was experimentally confirmed that the parameter $\beta_{2}$ for preserving the overall structure did not affect the results. Additionally, it was verified that the spatial flatness of the feature map of the illuminance component increased as γ increased. Figure 4 shows an example of the variation in the feature map for various $\beta_{1}$ and γ.

## 3. Experimental Results

### 3.1. Experimental Setup

Several experiments were conducted using various low-contrast images. The dataset used for training consisted of a pair of ground-truth and low-light images. Overall, 1300 ground-truth images were selected from the LIVE [32], Google Image-net [33], NASA ImageSet [34], and BSDS500 [35] datasets. The degraded images were generated from the ground-truth images using two random variables. The random variables were as follows: (1)gamma correction: $\Gamma \in \left[2.5,\;3.0\right]$,(2)random spray Gaussian noise: random spray ratio (0.01%) and Gaussian std. $\in \left[35.0,\;45.0\right]$.

A total of 6500 degraded images using the variables were randomly generated, and the average spatial resolution of the images was 884×654. In this work, we describe the experimental results of 50 real low-light images and distorted images of 50 ground-truth images that were not used for training. In addition, the parameters used in the loss function were set as $\alpha_{1}=\alpha_{2}=1.0$, $\beta_{1}=\beta_{2}=0.1$, and γ=0.01.

The proposed method compares the performance with MSR-net [18], Retienx-net [20], MBLLEN [21], and KinD [22] in terms of various evaluations, such as the peak-to-signal ratio (PSNR), lightness order error (LOE) [36], visual information facility (VIF) [37], perceptual based image quality evaluator (PIQE) [38], structural similarity index measure (SSIM) [39], and contrast per pixel (CPP) [40]. The LOE represents the number of pixels in which the lightness alignment between the reference image and the comparison image within a 50×50 window at the reference point deviates. The VIF is an evaluator that determines the degree of improvement or inhibition compared to the reference image using a statistically established index. In addition, PIQE, which does not require a reference image, represents the degree of natural representation of the image, where a smaller value indicates that the image is more natural from a cognitive perspective. On the other hand, CPP represents the amount of change in contrast within a 3×3 window, such that there is a limit to the evaluation of image quality improvement. However, it is suitable for evaluating the similarity of the amount of contrast change with the ground-truth image. An Intel E3-1276 3.6 GHz with 32 GB RAM and NVIDIA 1660Ti GPU were used to run the algorithms with the TensorFlow 1.2 library of Python 3.0. 

### 3.2. Analyses of Experimental Results

Table 1 shows the performance comparisons, where ↑ indicates a quantitative improvement as the value increases. The results show that the proposed method outperforms other methods in terms of PSNR, VIF, and PIQE. In particular, the PSNR, SSIM, VIF, and PIQE improved by 1.7~6.2 dB, 0.02~0.13, 0.04~0.2, and 7~10 with respect to the comparative methods, respectively. In contrast, MBLLEN dominates the others with respect to the LOE. Because the LOE evaluates the match between the alignment of the reference image and the corresponding comparative image as on/off, there is a limit to the accuracy of evaluating the performance improvement. In addition, the retinex-net generated halo-artifact, which is one of the main problems of retinex-based methods; further, it was confirmed that this distortion was a factor that increased the CPP value. In addition, it was observed that KinD outperforms the other comparative methods with respect to the PSNR, SSIM, and VIF. However, the LOE is very high due to the halo artifact. Through the results of the quantitative evaluations, it was confirmed that the proposed method reconstructed the image closest to the ground-truth image, and similar results were confirmed for the low-light image without the ground-truth image. In particular, the PIQE comparisons show that the proposed method has the capability to reconstruct more natural images.

Visual performance comparisons are shown in Figure 5 and Figure 6. It was observed that MSR-net is not promising with respect to luminance correction and color maintenance because it uses only feedforward training. In addition, the retinex-net has the problem of halo artifacts and color distortion. The halo artifact of the retinex-net is the main factor that increases the LOE and CPP, which agrees with the results shown in Table 1. The results verify that a different structure is applied to each channel due to insufficient correlation between RBG channels. On the other hand, MBLLEN is effective in removing additive noise using the convolutional neural layer based on an auto-encoder structure, but it is confirmed that illuminance improvement is insufficient and over-blurring is caused by over-denoising. KinD resulted in satisfactory illuminance improvement but led to color distortion and halo artifacts in the reconstructed images. However, the proposed method resulted in promising improvements. In particular, the experimental results show that the proposed method more naturally reconstructs the image compared with the other methods through illuminance improvement and color correction. As shown in Figure 6, similar results were obtained with real low-light images having uneven brightness and multi-light sources. Through the experimental results, it is confirmed that brightness correction, color maintenance, noise suppression, and halo artifact reduction should be simultaneously considered in low-light image enhancement. The experiments proved that the hybrid deep-learning structure and mixed-norm loss functions yield subjectively and objectively promising results.

## 4. Conclusions

This study introduces a hybrid deep-learning network and mixed norm loss functions, in which the hybrid net consists of a decomposition-net, illuminance enhance-net, and chroma-net, each of which is defined to reflect the properties. To improve brightness and reduce halo artifacts and color distortion, the YCbCr channels are used as inputs for the hybrid network. Then, the illuminance and reflectance are decoupled from the luminance channel, and the reflectance is trained by decomposition-net, such that the reflectance is enhanced, and the additive noise is efficiently removed. In addition, an enhance-net connected to the decomposition-net is introduced, resulting in illuminance improvement and reduction in halo artifacts. Moreover, the chroma-net is separately included in the hybrid-net because the properties of chroma channels are different from those of luminance, leading to a reduction in color distortion. In addition, a mixed norm loss function is introduced to minimize the instability and degradation of the reconstructed images by reflecting the properties of the reflectance, illuminance, and chroma.

The experiments confirmed that the proposed method showed satisfactory performance in various quantitative evaluations compared with other competitive deep-learning methods. In particular, it was verified that the proposed method could effectively enhance brightness and reduce additive noise, color distortion, and halo artifacts. It is expected that the proposed method can be applied to various intelligent imaging systems to obtain a high-quality image. Currently, deep-learning methods for low-light videos are under development. The newest methods are expected to reduce flickering artifacts between frames and to achieve even better performance.

## Figures and Tables

**Figure 1 sensors-22-06904-f001:**
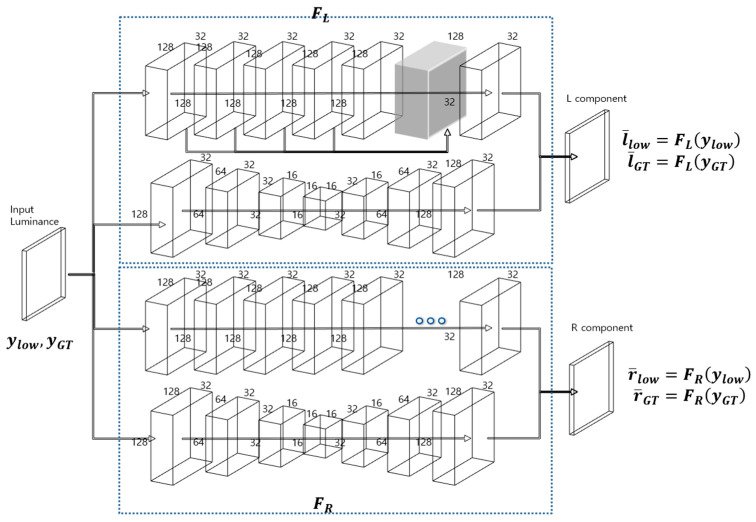
Conceptual diagram of proposed decomposition network.

**Figure 2 sensors-22-06904-f002:**
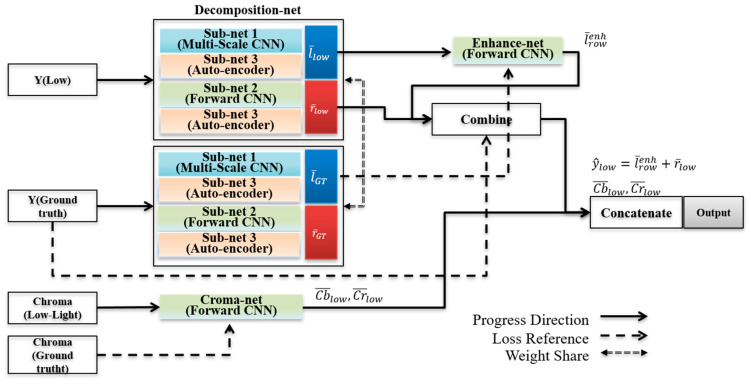
Flowchart of proposed network.

**Figure 3 sensors-22-06904-f003:**
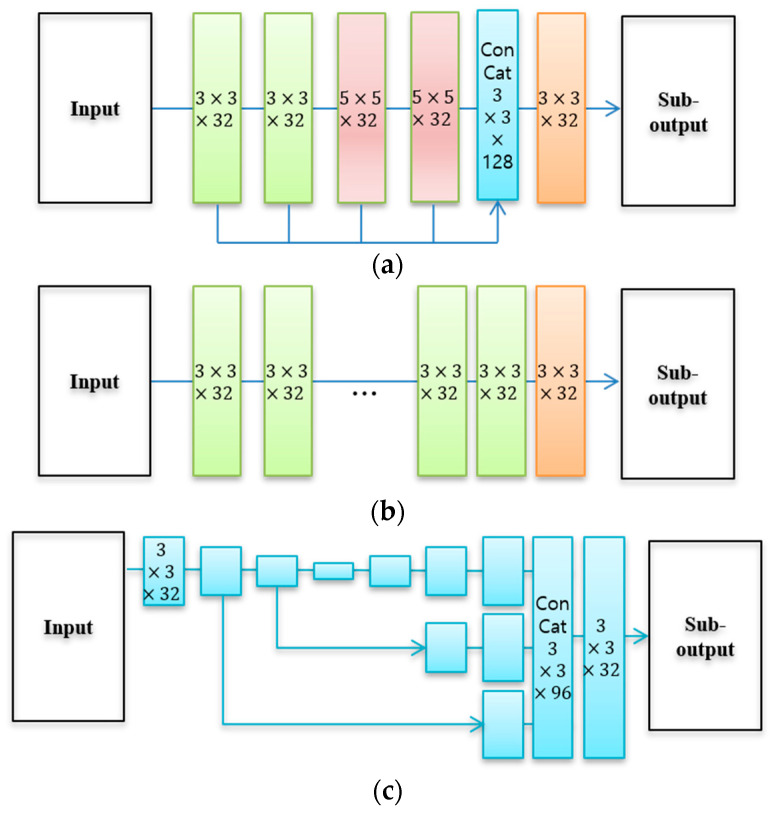
Architectures of sub-network: (**a**) multi-scale CNN (sub-net 1), (**b**) forward CNN (sub-net 2), (**c**) auto-encoder (sub-net 3).

**Figure 4 sensors-22-06904-f004:**
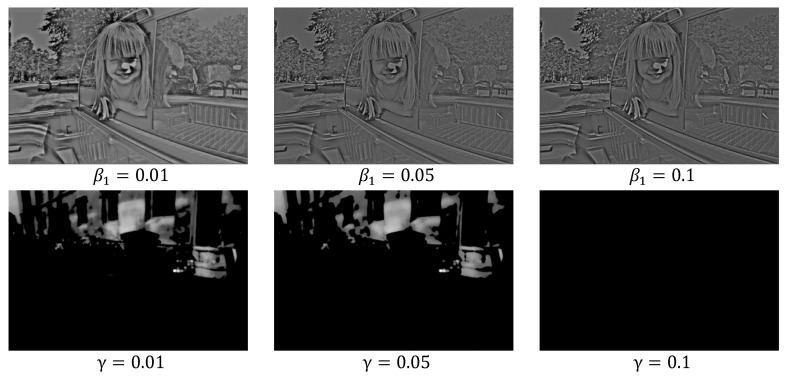
Variation of feature map for various $\beta_{1}$ and γ.

**Figure 5 sensors-22-06904-f005:**
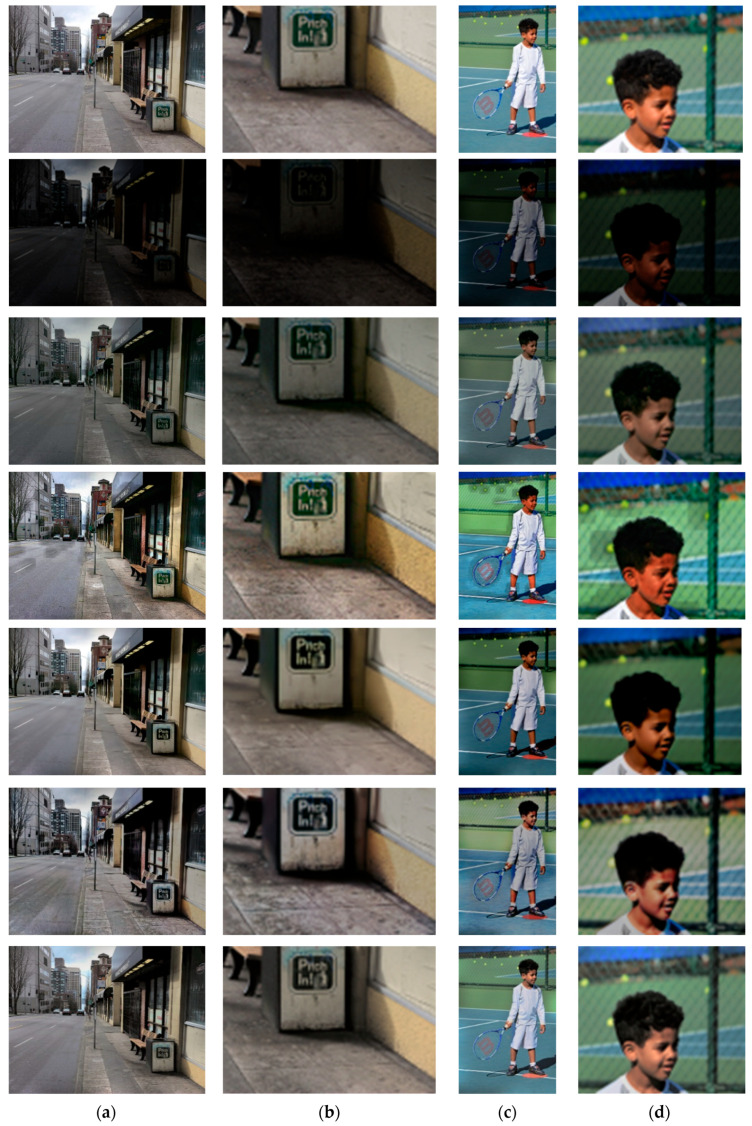
Visual comparisons of with-ground-truth images: (from top to bottom) ground-truth image, degraded image, MSR-net, retinex-net, MBLLEN, KinD, and proposed method. (**a**) test image1, (**b**) partially zoomed-in view of (**a**), (**c**) test image2, and (**d**) partially zoomed-in view of (**c**).

**Figure 6 sensors-22-06904-f006:**
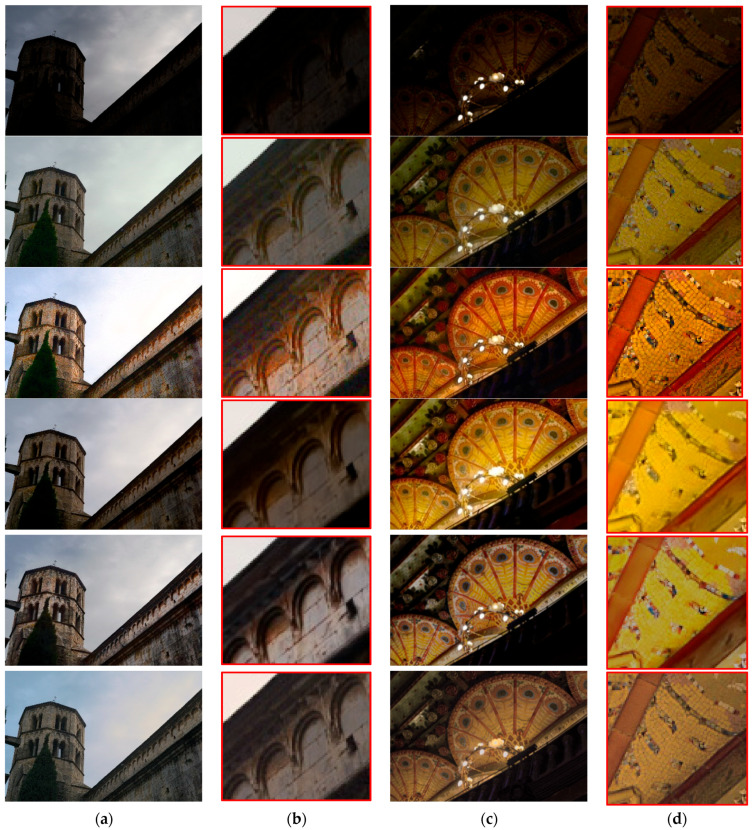
Visual comparisons of without-ground-truth images: (from top to bottom) real low-light image, MSR-net, retinex-net, MBLLEN, KinD, and proposed method. (**a**) test image3, (**b**) partially zoomed-in view of (**a**), (**c**) test image4, and (**d**) partially zoomed-in view of (**c**).

**Table 1 sensors-22-06904-t001:** Performance comparisons (**Blue**: the best, **Red**: the second best).

	Evaluator	Ground Truth	Degraded Image	MSR- Net [18]	Retinex- Net [20]	MBLLEN [21]	KinD [22]	Proposed Method
with reference	PSNR ↑	N/A	8.69	15.88	17.64	19.60	** 20.14 **	** 22.01 **
	SSIM ↑	N/A	0.547	0.800	0.766	0.823	** 0.873 **	** 0.897 **
	LOE ↓	N/A	282.90	210.94	374.09	** 202.57 **	327.84	** 208.04 **
	VIF ↑	N/A	0.366	0.508	0.451	0.556	** 0.613 **	** 0.656 **
	PIQE ↓	36.94	39.83	** 37.60 **	47.47	47.41	51.17	** 30.96 **
	CPP	35.98	15.07	29.36	** 47.50 **	25.82	30.03	** 31.27 **
without reference	PIQE ↓	N/A	33.25	** 31.06 **	39.25	52.65	46.17	** 24.02 **
	CPP	N/A	13.93	19.64	** 35.66 **	14.44	20.01	** 20.63 **

## Data Availability

Anyone can use or modify the source-code for only academic purposes. The source-code will be accessed on 15 October 2022 at https://drive.google.com/drive/folders/153qbJeMO96qSLS6qVr513v7_O8aIuKHI.

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
