# Peer review of "Low-Light Image Enhancement Using Hybrid Deep-Learning and Mixed-Norm Loss Functions"

_sensors, 2022, doi:10.3390/s22186904_

Round 1

Reviewer 1 Report

The article proposes an empirical model, machine-learning based, to improve low-light/under-exposed colour images.

The proposal is sound, the description is sufficient (but perhaps like many learning methods, results would be difficult to reproduce, unless the authors share the code). The experimentation is relevant, yet it is little in the explanability of the results (e.g. explainable networks). 

I think this is a clean piece of work and that it is publishable as it is.

Reviewer 2 Report

The paper describes low-light image enhancement method using a hybrid deep-learning network and mixed-norm loss functions. It consists of a decomposition-net, illuminance enhance-net and chroma-net. Claims include better results than state-of-the-art deep learning methods.

The paper is mostly well written, with good structure and the authors have outlined clearly their unique contributions.

However, I have some comments.

There are no mentions of GAN-based methods and others like LightNet. These should be added, where applicable. Also, images with actual low or uneven illumination should also be used to test for real-world conditions since those used here are artificially degraded images used for testing. This will show to some extent how suitable the proposed model is to resolving actual illumination degradation in real images.

Additionally, the EME and SSIM metrics need to be added for datasets with ground truths to assess the level of improvement too for benchmarking and comparison with the state-of-the-art algorithms.

Please correct this on line 279 to 281:

5. Conclusions

This section is not mandatory but can be added to the manuscript if the discussion is 280 unusually long or complex.

There are 34 references.

Two from 2019. 

One from 2020.

None from 2021.

One from 2022.

Need to update references for the last 3 to 4 years since a great deal of research papers have been published on similar topics as this in the (recent) space of this time.
